# Identification of a Novel Oligosaccharide in Maple Syrup as a Potential Alternative Saccharide for Diabetes Mellitus Patients

**DOI:** 10.3390/ijms20205041

**Published:** 2019-10-11

**Authors:** Kanta Sato, Noriaki Nagai, Tetsushi Yamamoto, Kuniko Mitamura, Atsushi Taga

**Affiliations:** Faculty of Pharmacy, Kindai University, 3-4-1 Kowakae, Higashi-Osaka, Osaka 577-8502, Japan; 1645110002y@kindai.ac.jp (K.S.); nagai_n@phar.kindai.ac.jp (N.N.); yamatetsu@phar.kindai.ac.jp (T.Y.); mitamura@phar.kindai.ac.jp (K.M.)

**Keywords:** oligosaccharide, maple syrup, invertase inhibitor, plasma glucose, diabetes mellitus

## Abstract

The incidence of diabetes mellitus (DM) is increasing rapidly and is associated with changes in dietary habits. Although restrictions in the use of sweeteners may prevent the development of DM, this might reduce the quality of life of patients with DM. Therefore, there has been a great deal of research into alternative sweeteners. In the search for such sweeteners, we analyzed the carbohydrate content of maple syrup and identified a novel oligosaccharide composed of fructose and glucose, linked at the C-4 of glucose and the C-6 of fructose. This oligosaccharide inhibited the release of fructose from sucrose by invertase (IC_50_: 1.17 mmol/L) and the decomposition of maltose by α-(1-4) glucosidase (IC_50_: 1.72 mmol/L). In addition, when orally administered together with sucrose to rats with DM, the subsequent plasma glucose concentrations were significantly lower than if the rats had been administered sucrose alone, without having any effect on the insulin concentration. These findings suggest that this novel oligosaccharide might represent a useful alternative sweetener for inclusion in the diet of patients with DM and may also have therapeutic benefits.

## 1. Introduction

A steady increase in the prevalence of metabolic syndrome can be ascribed to changes in dietary habits. This is becoming a serious social problem as it is associated with a higher risk of developing lifestyle-related diseases, including type 2 diabetes mellitus (DM), hypertension, and cardiovascular disease [1,2]. In particular, the prevalence of DM is rapidly increasing worldwide, and according to the International Diabetes Association, the number of people with DM will increase to 591.9 million by 2035 unless effective measures are taken. Restrictions on the use of sweeteners, such as sucrose, glucose, and fructose syrup are essential to prevent the development of DM and metabolic syndrome. However, artificial sweeteners that have been used as a substitute for sucrose have also been reported to increase the risk of DM [3]. Therefore, it is important to identify novel and safe alternative sweeteners that could improve the quality of life of type 2 DM patients.

Maple syrup is a popular natural sweetener that is consumed worldwide. Although sucrose constitutes ~60% of maple syrup, it also contains smaller concentrations of glucose and fructose [4,5,6,7,8]. However, Taga et al. has shown that maple syrup contains rare reducing carbohydrates such as xylose, arabinose, ribose, and mannose, using capillary electrophoresis (CE) analysis of 1-phenyl-3-methyl-5-pyrazolone (PMP) derivatives [9]. Furthermore, this study also suggested the presence of unidentified compounds that were considered to be oligosaccharides on the basis of their migration.

At present, various oligosaccharides are attracting attention and their functions have been elucidated. In addition, oligosaccharides have been reported to be effective against DM [10,11]. Therefore, it is possible that the unidentified oligosaccharides in maple syrup will be useful for the treatment of DM patients. However, the PMP derivatization that was previously used for the analysis of saccharides in maple syrup involves the conversion of terminal reducing sugars to potent ultraviolet-absorbing compounds [12,13] and it is not capable of detecting saccharides without a reducing terminal sugar. Thus, there may be other unidentified saccharides, which do not have terminal reducing sugars, that were not detected in the previous analysis.

In this study, we aimed to identify saccharide components of maple syrup after invertase digestion to remove fructose, including those with reducing terminal oligosaccharides, to determine whether any had been missed in previous analyses. We successfully identified a novel oligosaccharide in maple syrup that strongly interacts with invertase. Furthermore, we examined what effect this novel oligosaccharide has on the other glycosidases. Therefore, we hypothesize that this novel oligosaccharide has inhibitory activity against glycosidases which, in turn, has beneficial effects on hyperglycemia in DM animal models.

## 2. Results

### 2.1. Profile and Characteristics of the Saccharides in Maple Syrup

To profile the saccharides present in maple syrup, we performed CE separation of PMP-derivatized maple syrup components (Figure 1a), which yielded several peaks. Glucose, xylose, arabinose, and mannose were easily assigned on the basis of our previous findings [9], and ribose was identified to be the shoulder peak on that of glucose. In addition, there were several unassigned peaks that were considered to represent oligosaccharides on the basis of their migration time of ~20 min. Glucose was the most abundant of the detected saccharides.

Maple syrup samples were then digested with invertase and then PMP-derivatized to determine whether there were any oligosaccharides present that had remained unidentified due to capping of the reducing terminal by fructose (Figure 1b). The peak areas of glucose, xylose, and one of the unidentified oligosaccharides were larger than before invertase digestion, and melibiose was also identified (Figure 1b). However, the arabinose and ribose peaks disappeared because they overlapped with the large glucose peak. Here, oligosaccharides possessing fructose may be present in maple syrup, but obtaining standard samples was difficult. We considered the possibility of assigning saccharides possessing fructose by using lectin-recognizing carbohydrate residue. However, fructose-specific lectin could not be purchased on the general reagent market. Therefore, we tried using enzyme invertase for this purpose. To determine whether unidentified oligosaccharides may interact with invertase, invertase solution was added to PMP derivatives. As a result, the peak representing the unidentified oligosaccharide disappeared whereas the other peaks were unaffected by the addition of extra invertase after the original digestion (Figure 1c).

### 2.2. Purification and Structural Analysis of the Novel Oligosaccharides in Maple Sap

To analyze the structure of the unidentified oligosaccharides, maple sap was ultra-filtered to remove large molecules. We next characterized the saccharide components by HPLC, which is capable of detecting carbohydrates without derivatization. Although the sucrose, fructose, and glucose peaks could be assigned using standards, the other peaks, including that representing the invertase-interacting oligosaccharide could not be assigned (Figure 2a). Therefore, the fractions yielding unknown peaks were each further fractionated. Figure 2b shows the chromatogram generated while purifying the target oligosaccharide. This fraction was analyzed as a PMP derivative in order to assign a peak to the targeted oligosaccharide.

This fraction was acid-hydrolyzed to determine the monosaccharide composition of this oligosaccharide. The HPLC profile contained two major peaks, corresponding to glucose and fructose (Figure 2c), and the peak areas were almost equivalent. This implies that the oligosaccharide was likely to be a disaccharide composed of glucose and fructose. In addition, the oligosaccharide, derivatized using PMP, was analyzed by LC-ESI-MS/MS to confirm that it was a disaccharide. The peak associated with the PMP derivative had an *m/z* of 673.26 as [M + H]^+^ which corresponded with the predicted mass of the PMP-disaccharide. A product ion was observed at *m/z* 511.33, which was consistent with the predicted mass of the PMP-derivatized monosaccharide.

Subsequently, this oligosaccharide was subjected to NMR analysis to determine its structure more precisely. The proton and carbon signals collected are shown in Table 1. From the ^1^H-^13^C HMBC data, the structural formula of this oligosaccharide was predicted as shown in Figure 3. We called this novel oligosaccharide ‘maplebiose1′, and its function was then studied.

### 2.3. The Inhibitory Effect of Maplebiose1 on Enzymatic Glycolysis

Next, we determined whether maplebiose1 could inhibit glycolysis by interacting with invertase. Maplebiose1 inhibited invertase in a concentration-dependent manner, with the inhibition levels associated with 1 μg, 10 μg, and 100 μg of maplebiose1 being 40.3%, 43.6%, and 65.2%, respectively (Table 2). The relationship between maplebiose1 concentration and the level of inhibition was linear (R^2^ = 0.9984) and the IC_50_ was calculated to be 1.17 mmol/L.

We then determined whether maplebiose1 would have an inhibitory effect on the other glucosidases using rat intestinal acetone powder as a glucosidase mixture. The inhibition levels associated with maplebiose1 with respect to the metabolism of sucrose, maltose, and isomaltose were 12.3%, 9.4%, and 3.3%, respectively (Figure 4). Then, to confirm this inhibitory effect of maplebiose1 on maltose and isomaltose metabolism, IC_50_ values were determined using maltase and isomaltase. Maplebiose1 inhibited the activity of maltase with an IC_50_ of 1.72 mmol/L (Table 2), but the IC_50_ for isomaltose could not be determined because of its limited activity.

### 2.4. Prevention of the Hyperglycemia Induced in Rats Following Oral Administration of a Sucrose Load by Maplebiose1

Figure 5 shows the changes in plasma glucose (Figure 5a,b) and insulin (Figure 5c,d) concentrations in OLETF rats after oral administration of sucrose ± maplebiose1. Hyperglycemia and hypoinsulinemia were present in the 60-week-old OLETF rats, and the plasma glucose, TG, Cho, and insulin concentrations were 271 ± 5 mg/dL, 358 ± 13 mg/dL, 269 ± 14 mg/dL, and 59 ± 3 ng/dL, respectively (mean ± S.E.; *n* = 3). The plasma glucose concentrations peaked 60 min after the oral administration of sucrose solution and had not returned to the pre-administration concentrations 3 h after the sucrose administration. However, the hyperglycemia in the maplebiose1-treated rats was significantly less pronounced than that of the control rats after the oral administration of sucrose, and *AUC*_PG_ of the maplebiose1-treated rats was ~50% of that of the control rats. On the other hand, in insulin concentrations in OLETF rats, plasma insulin levels were maintained from 15 min after oral administration in both the control and maplebiose1-treated rats, and the *AUC*_insulin_ also did not differ between the control and maplebiose1-treated rats. This is a characteristic of OLETF rats with low insulin secretion [14]. As a result, it seems that the insulin levels were maintained by acting on glucose with a small amount of insulin secretion.

## 3. Discussion

There is a requirement to restrict the use of sweeteners, such as sucrose and honey, in patients with DM, but such restriction reduces their quality of life. Therefore, the identification of useful alternative sweeteners is likely to be of great interest. In the present study, we have carefully analyzed the carbohydrate composition of maple syrup and found several unidentified saccharides that might have properties suitable for this purpose. In particular, we have identified a novel oligosaccharide in maple syrup that interacts with invertase (Figure 1a–c). Therefore, we aimed to isolate and purify this saccharide to determine its structure and function. Because maple syrup is produced by boiling down maple sap, it is not suitable for the isolation of its components due to its high viscosity. Therefore, we successfully attempted to purify the target oligosaccharide from maple sap, in which it was present at a relatively high concentration. Structural analysis using NMR spectroscopy showed that the target oligosaccharide is a disaccharide composed of fructose and glucose (Figure 2b) with the C-4 of glucose linked to the C-6 of fructose (Figure 3). To our knowledge, this oligosaccharide is a novel saccharide that has a type of glycosidic bond not found in other disaccharides. We have designated this novel oligosaccharide ‘maplebiose1′.

We found that maplebiose1 interacts with invertase and prevents the digestion of sucrose by invertase in a concentration-dependent manner (Table 2). This finding led us to hypothesize that maplebiose1 might be able to inhibit the activity of other glucosidases. It is known that there are two types of enzyme that hydrolyze sucrose: invertase and sucrase. Invertase, also known as β-fructofuranosidase, cleaves the glycosidic bond by binding to sucrose from the fructose side, whereas sucrase binds to sucrose from the glucose side. Sucrase inhibitors such as acarbose and voglibose are already used therapeutically in DM patients. In addition, a number of compounds that have inhibitory effects on sucrase have been identified in natural materials, such as green tea [15,16,17]. In contrast, only a few studies have shown inhibitory effects of compounds on invertase [18,19,20]. Therefore, maplebiose1 might represent a rare compound that could be valuable in the management of DM. In addition, maplebiose1 is likely to be a relatively stable compound because it is present in maple syrup, which is produced by boiling down maple sap. In contrast, the previously discovered inhibitors were reported to be polymers, which are likely to be more heat sensitive.

Moreover, maplebiose1 had inhibitory effects on maltase and isomaltase, as components of rat intestinal acetone powder (Figure 4), which we attempted to confirm using purified enzyme preparations. Although an inhibitory effect on maltase was confirmed, limited inhibition of isomaltase was shown (Table 2). Previous studies have shown that the activity of isomaltase is enhanced by the formation of a complex in the small intestinal brush border [21,22]. Therefore, the conditions used in this analysis might not have been optimal for evaluation of the inhibitory effect of maplebiose1 on isomaltase. However, the IC_50_s of maplebiose1 for the inhibition of invertase and maltase were similar. Therefore, maplebiose1 is considered to be a useful biomaterial that can inhibit the digestion of both maltose and sucrose.

Subsequently, to confirm the effect of maplebiose1 in vivo experiments, we performed a sucrose intolerance tests using OLETF rats. OLETF rats, which have been established as a model of human type 2 diabetes mellitus are used for the genetic analysis of diabetes and development of therapeutic drugs [14,23,24,25]. In sucrose intolerance tests using OLETF rats, we have shown that maplebiose1 ameliorates sucrose intolerance, probably by restricting glucose absorption in the intestine. OLETF rats of >60 weeks of age are known to suffer β-cell loss in the pancreas, resulting in a reduction in the secretion of insulin [26]. Therefore, the plasma insulin concentrations and *AUC*_insulin_ of 60-week-old OLETF rats in this study were low, which is a result that was also shown in a previous study [24,25]. These results were accompanied by hyperglycemia, consistent with the development of DM. The increase in plasma glucose in rats administered with sucrose solution was significantly ameliorated by the co-administration of maplebiose1 (Figure 5). Although hyperglycemia is normally corrected by insulin secretion, the plasma insulin concentrations of the OLETF rats were not affected by the treatment. This suggested that invertase inhibition by maplebiose1 reduced the plasma glucose concentration. However, the increase of the plasma glucose concentration in OLETF rats was more strongly inhibited than predicted from the results of glycosidase inhibition in vitro. Therefore, it is suggested that maplebiose1 might possess another function for inhibition of the absorption of saccharides. In addition, we have previously reported that the increase in plasma glucose after oral administration of maple syrup to rats with DM is significantly lower than that of sucrose because of an inhibition of glucose absorption from the intestine [24,25,26]. Therefore, maplebise1 may be related to not only invertase inhibition but also glucose absorption in the small intestine. The present results contribute to our understanding of the observed health benefits and biological activities of maple syrup. In addition, maplebiose1 inhibits not only invertase, but also maltase. Maltose is produced by the digestion of starch, a grain-derived carbohydrate that is routinely consumed by people worldwide. Therefore, maplebiose1 might be useful for the prevention of the increase in plasma glucose after a meal. However, further studies are necessary to confirm the additional effects and metabolism that result from maplebiose1 administration.

In addition, it is necessary to examine the effect of maplebiose1 on DM patients in future studies, since these results are obtained by using diabetic-model rats. Although maplebiose1 is an ingredient in maple syrup which has been used for a long time worldwide, it is necessary to examine the long-term and side effects of maplebiose1 in humans who may use it in high concentrations, such as in a supplement.

In conclusion, we have isolated maplebiose1, a novel saccharide, from maple syrup and found that it is an invertase inhibitor in the present study. These findings support our hypothesis that there are beneficial effects associated with glycosidase on hyperglycemia.

## 4. Material and Methods

### 4.1. Materials and Animals

Chemicals and reagents of the highest grade available were purchased as follows. 1-Phenyl-3-methyl-5-pyrazolone (PMP) was purchased from Tokyo Chemical Industry Co., Ltd. (Tokyo, Japan). Invertase from *Candida utilis* was purchased from the Seikagaku Biobusiness Corporation (Tokyo, Japan). Intestinal acetone powder from rat and maltase (α-(1-4) glucosidase) from *S**accharomyces* were purchased from Sigma-Aldrich Japan K.K. (Tokyo, Japan). Maple syrup and sap was a kind gift from Maple Farms Japan, Inc. (Osaka, Japan). All other chemicals and reagents were purchased from Nacalai Tesque, Inc. (Kyoto, Japan).

Male 60-week-old OLETF rats obtained from Otsuka Pharmaceutical Co., Ltd. were provided with a commercial diet (Clea CE-2, CLEA Japan, Inc., Tokyo, Japan), and housed under fluorescent light between 07:00 and 19:00 at 25 °C. All procedures were approved by the Committee for the Care and Use of Laboratory Animals of Kindai University Faculty of Pharmacy and were performed according to the guidelines of Kindai University Animal Experimentation Regulation. (Approval number: KAFP-25-001, 1 April 2013)

### 4.2. PMP Derivatization of Maple Syrup and Sap

Maple syrup and sap were derivatized using PMP, according to the protocol of Honda et al. [27]. Briefly, 50 μL of 0.3 mol/L sodium hydroxide and 50 μL of 0.5 mol/L PMP in methanol were added to 10 mg of maple syrup or 200 μL of maple sap and heated at 70 °C for 30 min. After neutralization by the addition of 0.3 mol/L hydrochloric acid, 100 μL of distilled water was added and the mixture was extracted three times with 200 μL of chloroform to obtain the PMP-derivatized compounds.

### 4.3. Capillary Electrophoresis (CE)

Capillary electrophoresis was carried out using an Agilent (Agilent Technologies, Waldbronn, Germany) three-dimensional capillary electrophoresis system (Model G1600A) equipped with a diode array ultraviolet (UV) detector. Samples were injected under pressure at 5 kPa for 4 s. Separation was performed on an untreated fused silica capillary column (total length: 58.5 cm; effective length: 50 cm; internal diameter: 50 µm; GL Science Inc., Tokyo, Japan). A background electrolyte (BGE) of 200 mmol/L borate buffer (pH 10.5) was prepared by adding sodium hydroxide to sodium tetraborate solution and adjusting the pH. CE analysis was performed in the constant voltage mode, at +15 kV. Before each analysis, the capillary was flushed with 0.5 mol/L sodium hydroxide for 1 min and BGE for 5 min. Analytes were detected at UV absorption 245 nm. The capillary was kept at 25 ± 0.1 °C.

### 4.4. Invertase Digestion of Maple Syrup

Invertase digestion was performed as described previously, with slight modifications [9,28]. The enzyme reaction was carried out by adding 50 µL of 10 U/mL invertase solution in 5 mmol/L acetate buffer (pH 4.5) to 10 mg of maple syrup, after which the mixture was incubated at 37 °C for 15 min, then heated in boiling water for 1 min to inactivate the enzyme. The invertase-digested samples were derivatized with PMP after vacuum drying and analyzed by CE. Invertase digestion was performed as three independent experiments.

### 4.5. Size Exclusion Chromatography

To remove the high-molecular-weight components, the maple sap was ultra-filtered using Ultracel 10 kDa ultrafiltration discs (Merck Millipore, Darmstadt, Germany). The filtrate then underwent gel filtration using Sephadex G-15 gel (GE Healthcare, Boston, MA, USA) on a column (1 m × 25 mm internal diameter; Bio-Rad Laboratories, Inc., Hercules, CA, USA) with water to obtain the oligosaccharide fraction. Each fraction, consisting of 200 drops, was collected using a Bio-Rad fraction collector Model 2110 (Bio-Rad Laboratories, Inc.).

### 4.6. High-Performance Liquid Chromatography with Charged Aerosol Detection (HPLC-CAD)

The HPLC system consisted of a Shimadzu (Kyoto, Japan) model LC-10AD pump, a Shimadzu degasser model DGU-12A, and a Corona Veo detector (Thermo Fisher Scientific, Inc., Waltham, MA, USA). To analyze the oligosaccharide fraction, an Asahipak NH2P-50 4E column (5 µm, 4.6 mm internal diameter ×250 mm, Showa Denko K.K., Tokyo, Japan) was used, and the mobile phase was acetonitrile/water (3:1; *v*/*v*). Elution was carried out at a flow rate of 1 mL/min at room temperature (~23 °C). The injection volume was 20 µL. To purify the target oligosaccharide, an Asahipak NH2P-50 10E column (5 µm, 10.0 mm internal diameter ×250 mm; Showa Denko K.K.) was used with a flow rate of 2 mL/min. As a post-column type splitter, an Adjustable Flow Splitter (Thermo Fisher Scientific Inc., Waltham, MA, USA) was used. The split ratio was 1:20, with the low-flow outlet directed to CAD. The remaining (~95%) volume was then collected from the high-flow outlet of the splitter. The other conditions were as described above.

### 4.7. Hydrolysis of the Unidentified Oligosaccharide

The purified oligosaccharide solution obtained was evaporated to remove the acetonitrile and lyophilized. One hundred micrograms of the dried sample were dissolved in 100 µL of 0.1 mol/L HCl and heated in a boiling water bath for 15 min. Then, a 20 µL aliquot of this solution was subjected to LC-CAD analysis.

### 4.8. Liquid Chromatography (LC)-Electrospray Ionization (ESI)-Mass Spectrometry (MS)

The LC-ESI-MS/MS system consisted of a Finnigan LTQ linear ion trap mass spectrometer (Thermo Fisher Scientific Inc.) coupled with an ESI interface, Paradigm MS4 pump (Michrom Bioresources Inc., Auburn, CA, USA) and an HTC PAL LC-autosampler (CTC Analytics GmbH, Zwingen, Switzerland). The parameters for analysis are shown below. The ion source voltage was set at 4.5 kV. The tube lens offset and capillary voltage were set at 90 V and 25 V, respectively. The capillary temperature was set at 275 °C. Nitrogen was used as sheath gas and auxiliary with 50 and 5 arbitrary units, respectively. Helium gas was used as the collision gas with a collisional activation amplitude of 35%. TSK gel ODS-100S (5 μm, 150 × 2.0 mm internal diameter, Tosoh Co., Tokyo, Japan) was used in reverse-phase mode. The mobile phase consisted of 0.1% formic acid (solvent A) and acetonitrile (solvent B) with linear gradient elution in 10–60% solvent B over 25 min at a flow rate of 200 μL/min. The injection volume was 10 µL.

### 4.9. Nuclear Magnetic Resonance (NMR)

^1^H and ^13^C-NMR spectroscopic data were obtained using a model JNM-ECA 800 (Jeol Resonance, Inc., Tokyo, Japan) instrument operating at 800 and 200 MHz, respectively. D_2_O was used as the deuterated NMR solvent. ^1^H-^1^H correlation spectroscopy (COSY), ^1^H-^13^C heteronuclear single-quantum correlation spectroscopy (HSQC), and ^1^H-^13^C heteronuclear multiple-bond correlation spectroscopy (HMBC) spectra were obtained using gradient-selected pulse sequences.

### 4.10. Invertase Inhibition Assay

Invertase inhibition assays were carried out as described in Section 4.4. The enzymatic reaction was inhibited by adding the novel oligosaccharide, named ‘maplebiose1′, concentrations of 1, 10, or 100 µg/50 µL in 100 mmol/L acetate buffer (pH 4.5) to 100 µg sucrose, and initiated by the addition of 50 µL of 0.2 U/mL invertase solution in water. After a 15-min incubation, 10 µL aliquots were collected from the reaction mixture and heated in boiling water for 10 min to inactivate the enzyme. The amount of glucose in this reaction mixture was then determined by CE after PMP derivatization. A blank sample was similarly treated, but in the absence of the inhibitor. The inhibition by maplebiose1 was calculated using the following equation.
(1)$$\mathrm{Inhibition}(\%)=[1-(\frac{D2-D3}{D1})]\times 100$$
where D1 is the glucose peak area of the blank sample, D2 is the glucose peak area of each sample with inhibitor, and D3 is the glucose peak area of each sample without substrate.

The half-maximal inhibitory concentration (IC_50_) value was calculated from the concentration-dependent inhibitory curve for maplebiose1.

### 4.11. Glycosidase Inhibition Assay

Glycosidase inhibition assay was performed according to a previous study, with slight modifications [29]. Briefly, to test glycosidase inhibitory effects, 3.4 mg of maplebiose1 was dissolved in 1 mL of 50 mmol/L phosphate buffer (pH 6.0). To prepare a crude glycosidase solution, 450 μL of 50 mmol/L phosphate buffer (pH 6.0) was added to 50 mg of rat intestinal acetone powder and then homogenized for 30 sec. After centrifugation at 9000 × *g* at 4 °C for 20 min, the supernatant was collected and used as a crude glycosidase solution. Each 100 µL of substrate solution, consisting of 100 µmol/L of maltose, isomaltose, or sucrose in phosphate buffer solution (pH 6.0) was mixed with 10 μL of the inhibitor solution, and pre-incubated at 37 °C for 5 min. Then, 90 μL of the crude enzyme solution was added to this mixture and incubated at 37 °C for 5 h. A 10 µL aliquot of the reaction solution was then collected and heated in a boiling water bath for 10 min to stop the reaction. The amount of glucose in the reaction mixture was determined by CE as a PMP derivative. A blank was also prepared, as for the invertase inhibition assay. The inhibition rate was calculated using Equation (1).

For maltase (α-(1-4) glucosidase) and isomaltase (α-(1-6) glucosidase) inhibition assays, 1, 10, or 100 µg of maplebiose1 were added to maltose or isomaltose in 50 µL of 10 mmol/L citric buffer (pH 6.8) or 100 mmol/L acetate buffer (pH 4.5). Then the reaction was started by the addition of 50 µL of 2 U/mL maltase or isomaltase solution in water. The subsequent procedure was then identical to that for the invertase inhibition assay. The inhibition of maltase and isomaltase was also calculated using Equation (1).

### 4.12. Biochemical Assays Conducted on Blood Samples from Rats

Measurements of blood parameters associated with diabetes mellitus were made as previously described [24,25]. Blood was drawn from a tail vein of rats fasted for 14 h at AM 10:00, and plasma was separated. Plasma glucose (PG) levels and triglycerides (TG) were determined by the Glucose Assay Kit (BioVision Inc, Milpitas, CA, USA) and Accutrend GCT (Roche Diagnostics, Mannheim, Germany). Total-cholesterol (Cho) levels were measured using a Cholesterol E-Test Kit (Wako, Osaka, Japan). Insulin levels were determined by an Insulin ELISA Kit (Morinaga Institute of Biological Science Inc., Kanagawa, Japan). All procedures were performed according to the manufacturers’ instructions.

### 4.13. Oral Sucrose Tolerance Testing

Sucrose (1.5 g/kg), with or without oligosaccharide (1.62 mg/kg), was administered orally to rats fasted for 14 h, and blood samples were taken from tail veins at subsequent time points for the measurement of plasma glucose (PG) and insulin concentrations. The differences in the plasma glucose (Δ*C*_PG_, mg/dL) and insulin (Δ*C*_insulin_, ng/dL) concentrations between rats that were or were not administered with sucrose were analyzed, and the areas under the plasma concentration–time (*t*, min) curves for PG (*AUC*_PG_) and insulin (*AUC*_insulin_) were calculated using the following equation.

(2)$$AUC_{\mathrm{PG}\;\mathrm{or}\;\mathrm{insulin}}\;\mathrm{in}\;\mathrm{OLETF}\;\mathrm{rat}=\int_{0\min}^{180\min}\Delta C_{\mathrm{PG}\;\mathrm{or}\;\mathrm{insulin}}\;dt$$

### 4.14. Statistical Analysis

All experiments were repeated at least three times. All experimental data are presented as the mean ± standard errors (SEs) of the mean. Statistical analyses were performed by Student’s or Aspin–Welch’s *t*-tests. *p* values less than 0.05 were considered to represent a statistically significant difference.

## Figures and Tables

**Figure 1 ijms-20-05041-f001:**
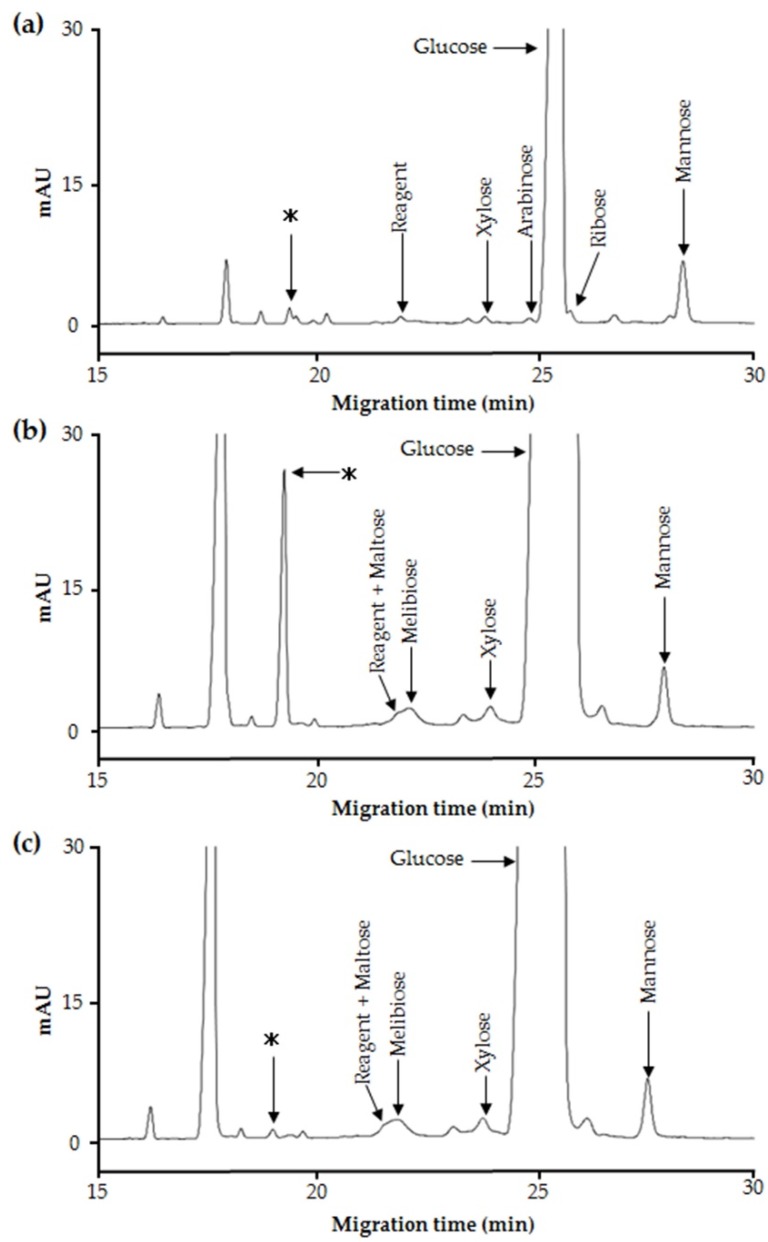
Pherograms of (**a**) the saccharide components of maple syrup, (**b**) the saccharide components of maple syrup after invertase digestion, and (**c**) the saccharide components of maple syrup after the addition of invertase to (**b**). The peak (*) was increased in size by the invertase digestion of maple syrup and disappeared after addition of further invertase. Analytical conditions: apparatus, Agilent G1600A; background electrolyte (BGE), 200 mmol/L borate buffer (pH 10.5); conditioning, 0.5 mol/L NaOH for 1 min, followed by BGE for 4 min; applied voltage, 15 kV; capillary, untreated fused silica (50 µm internal diameter, 58.5 cm, 50 cm); detection, UV absorption at 245 nm; temperature, 25 °C; sample introduction, hydrostatic method (5 kPa. 4.0 s).

**Figure 2 ijms-20-05041-f002:**
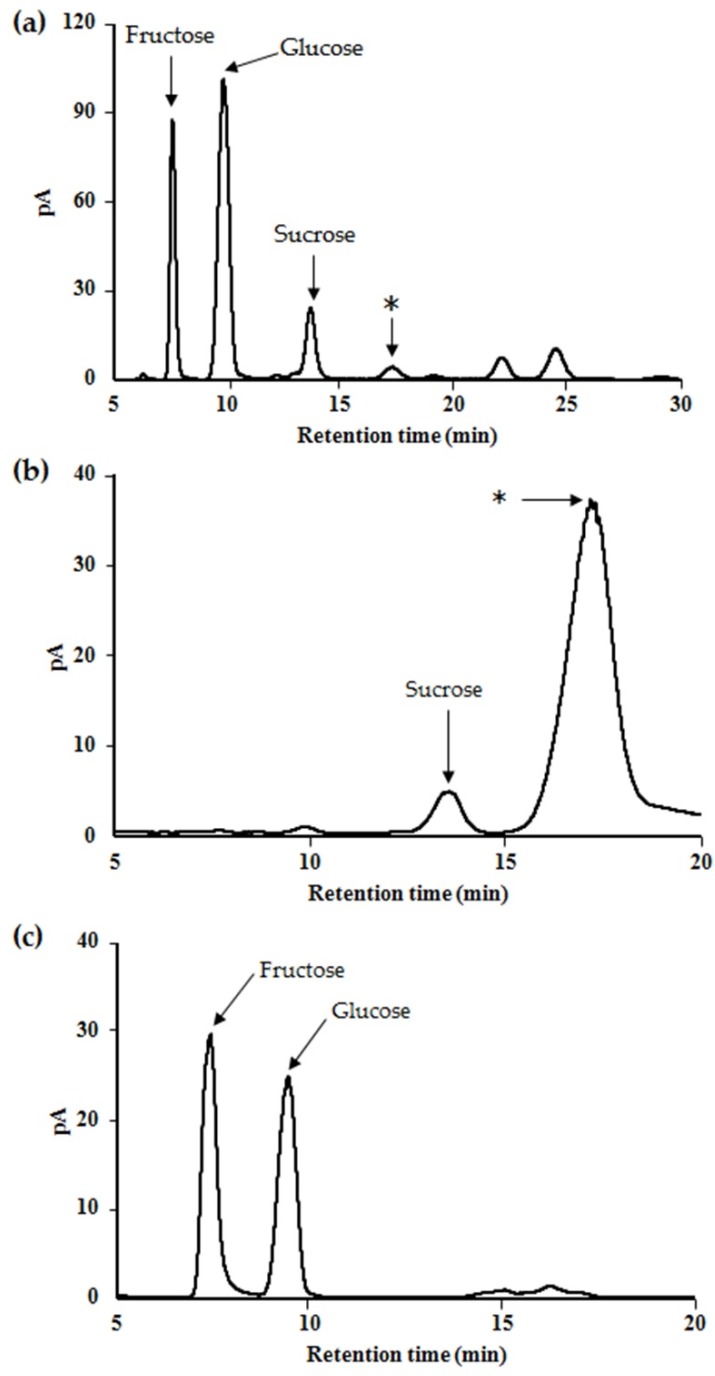
Chromatograms of (**a**) the saccharides components in maple sap after ultrafiltration, (**b**) the purified oligosaccharide, and (**c**) the saccharide components after the hydrolysis of the purified oligosaccharide. The peak (*) was obtained by fractionating and purifying an unidentified oligosaccharide from maple sap. Analytical conditions: apparatus, LC-10AD pump, degasser model DGU-12A and a Corona Veo detector; column, Asahipak NH2P-50 4E (5 µm, 4.6 mm internal diameter × 250 mm); mobile phase, acetonitrile/deionized water (3:1; *v*/*v*); flow rate, 1 mL/min; injection volume, 20 µL; column temperature, 25 °C.

**Figure 3 ijms-20-05041-f003:**
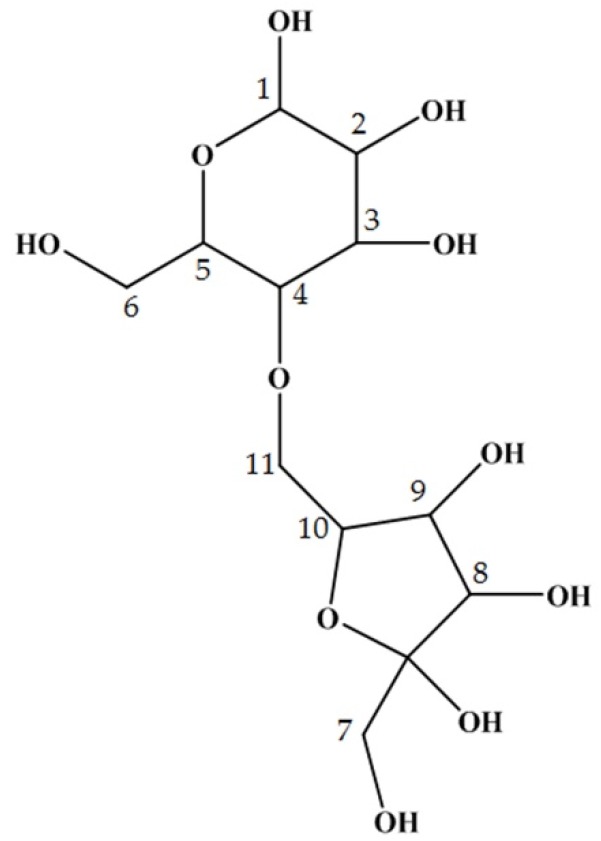
The structure of maplebiose1, obtained from maple sap. The proton and carbon signals for each number are listed in Table 1.

**Figure 4 ijms-20-05041-f004:**
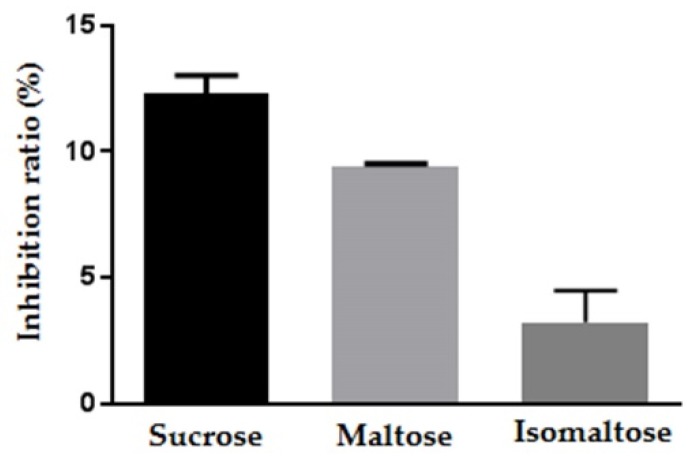
The inhibitory effect of maplebiose1 on α-glucosidase was identified using rat intestinal acetone powder. The results are presented as mean ± S.E.

**Figure 5 ijms-20-05041-f005:**
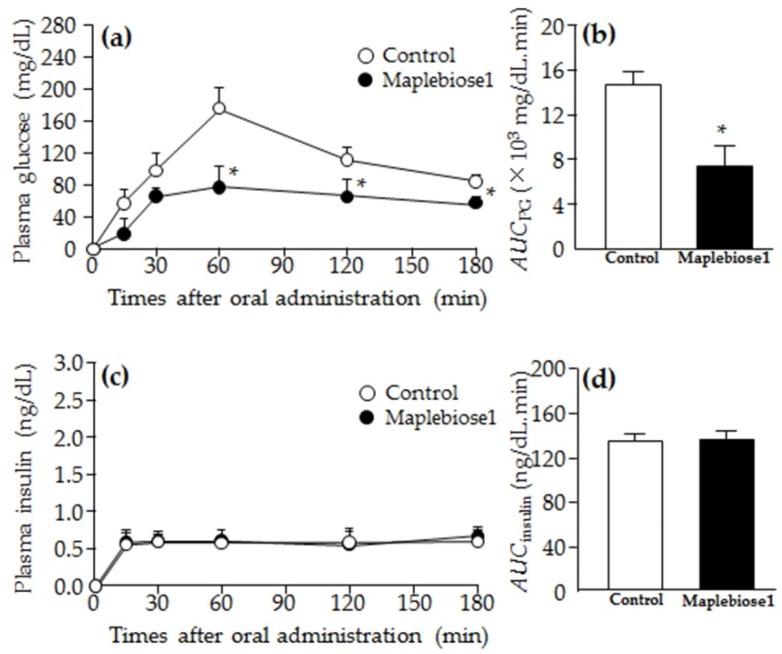
Effects of maplebiose1 on plasma glucose and insulin concentrations in OLETF rats administered an oral sucrose load. (**a**) Change in the plasma glucose concentration in OLETF rats following the oral administration of sucrose. (**b**) *AUC*_PG_ following the oral sucrose load. (**c**) Change in plasma insulin concentration following the oral sucrose load. (**d**) *AUC*_insulin_ following the oral sucrose load. Control, OLETF rats administered with sucrose; Maplebiose1, OLETF rats administered with sucrose and maplebiose1. The data are presented as means ± SEs for three rats. * *p* < 0.05 vs. Control for each category.

**Table 1 ijms-20-05041-t001:** ^1^H (800 Mz) and ^13^C (200 Mz) nuclear magnetic resonance spectral data for maplebiose1 in D_2_O.

Chemical Shift (ppm)			
Position	H (Proton)	C (Carbon13)	Intensity
1	4.4832	98.5740	57.4308
2	3.0938	76.6692	51.0309
3	3.3153	78.2276	64.6557
4	3.2818	72.2547	63.2886
5	3.3905	77.5813	39.4174
6	3.5403	62.7243	180.0717
7	3.5912	62.7459	58.2694
8	4.0197	79.5228	101.6335
9	3.9639	77.2054	62.9815
10	3.7169	83.8104	46.7190
11	3.8738	63.3779	34.1492

Chemical shifts are shown in ppm and coupling constants are shown in Hz in parentheses.

**Table 2 ijms-20-05041-t002:** Inhibitory effects of maplebiose1 on glycolytic enzymes.

Glycolytic Enzyme	Inhibitor (µg)	Peak Response (D2)	Peak Response (D3)	Inhibition Rate (%)	IC50 (mmol/L)
Invertase	-	30.5	-	-	1.17
	1	18.2	N.D	40.3	
	10	17.2	N.D	43.6	
	100	23.1	12.5	65.2	
Maltase	-	44.3	-	-	1.72
	1	27	N.D	39.1	
	10	22.4	N.D	49.4	
	100	20.3	N.D	54.2	
Isomaltase	-	31.6	-	-	-
	1	21.1	N.D	33.2	
	10	31.9	N.D	-	
	100	46.7	6.7	-	

D2: Glucose peak area for each sample; D3: Glucose peak area for each sample without substrate; N.D. not detectable; IC50: half-maximal inhibitory concentration.

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
