# Peer review of "Identification of a Novel Oligosaccharide in Maple Syrup as a Potential Alternative Saccharide for Diabetes Mellitus Patients"

_ijms, 2019, doi:10.3390/ijms20205041_

Round 1

Reviewer 1 Report

The overall premise of this manuscript is quite interesting and novel. The authors identified and isolated a novel oligosaccharide-maplebiose1, and tested its function in in vitro and in vivo. The experiments were designed and conducted properly. However several statements throughout the paper were unclear or need more evidence.

Lines 36-38 on page 2, which seems not related to the study. The statement “Although hyperglycemia is normally corrected…maplebiose1 may reduce glucose absorption” (Lines 18-20 on page 8) may not be comprehensive. According to their in vitro experiments maplebiose1 may also inhibit the breakdown of sucrose. Another concern is that the plasma insulin levels (Figure 5C) were stay high even 3 hours after administration. What is the explanation for this? The authors pointed out that maplebiose1 is a potential alternative saccharide for diabetic patients, so what is the long-term effect or is there any known side effect? I did not see the evidence for the statement “we have provided evidence that it prevents the absorption of glucose in the small intestine, using normal and diabetic rats” in conclusion.

Author Response

#Reviewer 1

Lines 36-38 on page 2, which seems not related to the study.

Thank you very much for pointing this out. We checked and delete these sentences.

 The statement “Although hyperglycemia is normally corrected…maplebiose1 may reduce glucose absorption” (Lines 18-20 on page 8) may not be comprehensive. According to their in vitro experiments maplebiose1 may also inhibit the breakdown of sucrose.

Thank you very much for your helpful comments. As suggested by a reviewer, we have not confirmed that maplebiose1 directly inhibit glucose absorption from the small intestine. However, the inhibitory effect of maplebiose1 on the increase of plasma glucose levels in vivo was stronger than that predicted from the in vitro glycosidase inhibitory effect. On the other hand, it has been reported that maple syrup reduced glucose absorption in small intestine (Ref. 25). Therefore, we considered that maplebiose1 might be involved in the intestinal absorption of glucose. In response to this comment, we deleted this sentence and added the above consideration in “Discussion” section (Lines 24-33 on page 8).

Another concern is that the plasma insulin levels (Figure 5C) were stay high even 3 hours after administration. What is the explanation for this?

Thank you very much for your helpful comments. OLETF rats used in this study are diabetic model rats with low insulin secretion and on the plasma insulin levels. In response to this comment, we added this information in “Results” section (Lines 19-22 on page 6, Lines 1-2 on page 7) and “Discussion” section (Lines 18-21 on page 8).

The authors pointed out that maplebiose1 is a potential alternative saccharide for diabetic patients, so what is the long-term effect or is there any known side effect?

Thank you very much for pointing this out. Maple syrup is a natural sweetener used for many people and for a long time. Therefore, we considered that maplebiose1 which contained maple syrup has less advance effect against human as compared to other diabetic medicine. However, it is important to examine the long-term effects of maplebiose1. Therefore, we will next investigate the long-term effect and side effect of maplebiose1 in diabetic patients. In response to this comment, we added the sentence about future experiments in “Discussion” section (Lines 39-43 on page 8).

I did not see the evidence for the statement “we have provided evidence that it prevents the absorption of glucose in the small intestine, using normal and diabetic rats” in conclusion.

Thank you very much for your helpful comment. According to your suggestion, we deleted this sentence.

Reviewer 2 Report

In the manuscript, "Identification of a novel oligosaccharide in maple syrup as a potential alternative saccharide for diabetes mellitus patients," Sato and colleagues have identified a novel oligosaccharide in maple syrup. While this is interesting concerns are raised throughout the manuscript.

In the background, the authors have focused on the effects of the oligosaccharides in a mouse model, but make no mention of the research in humans. Diabetes in mice/rats responds very differently than diabetes in humans.

In the methods, it appears that in the invertase digestion, the peak disappears after further digestion. This would seem to suggest that it was not completely digested the first time. 

The authors also comment on the "inhibitory" effect of maplebiose1, but the question arises, is the concentration of maplebiose1 in such an excess to other substrate that it appears to give the appearance of inhibition. In any system that has an excess of one substrate, it will be used.

In the oral sucrose tolerance test, the rats were not given maplebiose1 alone to look at the impact on glucose tolerance, only in combination with sucrose. Again while this could be inhibiting the conversion of sucrose to glucose and fructose, it is unclear what its isolated effect would be. Also the this would lead one to believe that their may be some issues with malabsorption. The authors make no comment on stooling in the animals.

A description and rationale for the biochemistry is needed.

Author Response

#Reviewer 2

In the background, the authors have focused on the effects of the oligosaccharides in a mouse model, but make no mention of the research in humans. Diabetes in mice/rats responds very differently than diabetes in humans.

Thank you very much for your helpful comment. As suggested by a reviewer, it is necessary to examine the effect of maplebiose1 on human DM patients to clarify its effect. However, the OLETF rat used in this study is considered as diabetic animal model which close to the enzyme system of human DM patients. Therefore, it is used for genetic analysis of diabetes and development of therapeutic drugs. In response to this comment, we added the information about usefulness of these animal model and the necessity of examination using DM patients as the future study in “Discussion” section (Lines 14-16, 39-43 on page 8).

 In the methods, it appears that in the invertase digestion, the peak disappears after further digestion. This would seem to suggest that it was not completely digested the first time. 

Thank you very much for your useful suggestion. As the reviewer suggested, there is a possibility that invertase digestion of maple syrup may be incomplete. However, we have optimized the invertase digestion conditions of maple syrup which gave reaction curves having plateau with appropriate reproducibility. Therefore, we thought invertase digestion of maple syrup might be complete. In response to this comment, we added information of the experiments in “Material and Methods” section (Lines 35-36 on page 9).

The authors also comment on the "inhibitory" effect of maplebiose1, but the question arises, is the concentration of maplebiose1 in such an excess to other substrate that it appears to give the appearance of inhibition. In any system that has an excess of one substrate, it will be used.

Thank you very much for your kindly comment. We performed enzymatic reactions according to a previous paper. In response to this comment, we added the information and the reference about enzymatic experiments in “Material and Method” section (Lines 2-3 on page 11).

 In the oral sucrose tolerance test, the rats were not given maplebiose1 alone to look at the impact on glucose tolerance, only in combination with sucrose. Again while this could be inhibiting the conversion of sucrose to glucose and fructose, it is unclear what its isolated effect would be. Also the this would lead one to believe that their may be some issues with malabsorption. The authors make no comment on stooling in the animals.

Thank you very much for your kindly comment. As suggested by a reviewer, we have not administered maplebiose1 alone to the rats. Therefore, we think that it is necessary to administer maplebiose1 alone to the rats for study of the effect and metabolism of maplebiose1 on small intestine of rat. In response to this comment, we added the information of our consideration about isolated effect of maplebiose1 and future work in “Discussion” section (Lines 37-38 on page 8).

Round 2

Reviewer 1 Report

My concerns were answered. But it might be better to review the text by a native English speaker.

Author Response

Thank you very much for your useful advice. As suggestion by a Reviewer, the revised version has been checked by MDPI English Editing service, we will upload the proof.

Reviewer 2 Report

The authors have made revisions to the manuscript to answer the reviewers comments; however, a few things still need to be addressed.

In the introduction, the authors still spend a significant amount of time discussing diabetes cures in animal models. This is not appropriate for the present manuscript.

The invertase digestion is still slightly confusing in that the peak disappears with additional invertase. As mentioned that this may be a result of an interaction with inveratase, but this really needs more thorough explanation.

The authors also mentioned that they would look at the effect of maplebiose1 in future experiments, but this really needs be be included in the current manuscript for be able to draw appropriate conclusions.

Author Response

#Reviewer 2

In the introduction, the authors still spend a significant amount of time discussing diabetes cures in animal models. This is not appropriate for the present manuscript.

Thank you very much for pointing this out. In response to this comment, we deleted the sentence and added the brief summarized sentence in “Introduction” section (Lines 1-2 on page 2).

 The invertase digestion is still slightly confusing in that the peak disappears with additional invertase. As mentioned that this may be a result of an interaction with inveratase, but this really needs more thorough explanation.

Thank you very much for your kind comment. In response to this comment, we deleted the sentence and added the explanation that it is an interaction between maplebiose1 and invertase in “Results” section (Lines 29-34 on page 2).

 The authors also mentioned that they would look at the effect of maplebiose1 in future experiments, but this really needs be be included in the current manuscript for be able to draw appropriate conclusions.

Thank you very much for your kind advices. We would like to do future work immediately. However, we cannot perform it at this time, because there is only very small amount of maplebiose1 in maple syrup and sap. It will take time to prepare. In response to this comment, we deleted the sentence in “Discussion” section.

Round 3

Reviewer 2 Report

The authors have provided sufficient response to the concerns.